# TAMING MASKED DIFFUSION LANGUAGE MODELS VIA CONSISTENCY TRAJECTORY REINFORCEMENT LEARNING WITH FEWER DECODING STEP

## ABSTRACT

Masked diffusion language models (MDLMs) have recently emerged as a promising alternative to autoregressive (AR) language models, offering properties such as parallel decoding, flexible generation orders, and the potential for fewer inference steps. Despite these advantages, decoding strategies and reinforcement learning (RL) algorithms tailored for MDLMs remain underexplored. A naive approach is to directly transfer techniques well-established for AR models to MDLMs. However, this raises an immediate question: Is such a naive transfer truly optimal? For example, 1) Block-wise and semi-AR decoding strategies are not employed during the training of MDLMs—so why do they outperform full diffusion-style decoding during inference? 2) Applying RL algorithms designed for AR models directly to MDLMs exhibits a training-inference inconsistency, since MDLM decoding are non-causal (parallel). This results in inconsistencies between the rollout trajectory and the optimization trajectory. To address these challenges, we propose **EOS Early Rejection** (**EOSER**) and **Ascending Step-Size** (**ASS**) decoding scheduler, which unlock the potential of MDLMs to perform full diffusion-style decoding, achieving competitive performance with fewer decoding steps. Additionally, we introduce **C**onsistency Tra**j**ectory **G**roup **R**elative **P**olicy **O**ptimization (**CJ-GRPO**) for taming MDLMs, which emphasizes the consistency between rollout trajectory and optimization trajectory, and reduces the optimization errors caused by skip-step optimization. We conduct extensive experiments on reasoning tasks, such as mathematical and planning benchmarks, using LLaDA-8B-Instruct. The results demonstrate that the proposed EOSER and ASS mechanisms, together with CJ-GRPO, hold significant promise for effectively and efficiently taming MDLMs.

## 1 INTRODUCTION

Autoregressive (AR) large language models Brown et al. (2020); Bai et al. (2023); Touvron et al. (2023); Grattafiori et al. (2024); Guo et al. (2025); Yang et al. (2025) have demonstrated compelling scaling laws in general capabilities, alignment, and reasoning. Their success suggests that scaling up the model size, training corpora and computational resources can yield models capable of solving a broad spectrum of tasks. However, beyond sheer scale, there is no evidence indicating that autoregressive modeling is the inevitable or optimal foundational paradigm for language modeling. Diffusion language models (DLMs) Khanna et al. (2025); DeepMind (2025); Song et al. (2025) have recently emerged as a promising competitor, exhibiting parallel decoding (non-causal, high throughput), flexible orders, and potential for fewer inference steps. Built upon prior theoretical and empirical work Austin et al. (2021); Lou et al. (2023); Nie et al. (2024), masked diffusion language models (MDLMs) have stood out from counterparts (i.e., DLMs) and established a dominant position.

A representative train-from-scratch MDLM—LLaDA Nie et al. (2025) demonstrates competitive performance compared to similarly sized AR models. Beyond the promising results, two problems regarding MDLMs remain underexplored: **(1) Decoding Strategy**, as shown in the (d) of Fig. 1, why does semi-AR (i.e., block-wise) decoding outperform full diffusion (without block constraint) decoding without specifically training for this strategy? The (a), (b) and (c) of Fig. 1 may unveil the reason behind this behavior. In particular, (c.1) and (c.2) indicate two key characteristics: 1) The

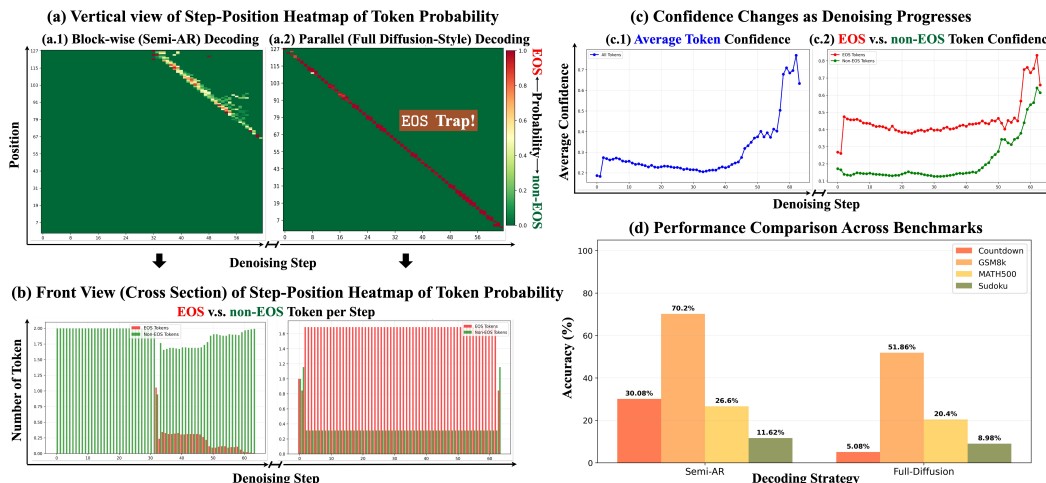

Figure 1: **(a)** and **(b)** illustrate the vertical and front view of Step-Position Heatmap of Token Probability, showing the frequency of `<EOS>` appearing at each position per step. **(c)** depicts the performance comparison of semi-AR and full-diffusion decoding strategies. **(d)** shows the average confidence of tokens as denoising progresses.

average tokens confidence is lower in the initial denoising steps but increases sharply as denoising progresses. 2) The average confidence of `<EOS>` is significantly higher than that of `<non-EOS>`. These characteristics explain the phenomena observed in (a) and (b). The block constraint avoids decoding the `<EOS>` at the end of sequence in early steps, whereas full diffusion decoding lacks such constraint. Consequently, when employing full diffusion decoding, few semantically informative tokens are decoded in the initial low-confidence phase, potentially causing derailment—the model deviates from the correct denoising trajectory. We refer to this phenomenon the '`<EOS>` Trap!' of full diffusion decoding. Furthermore, characteristic 1) suggests that uniform step-size decoding scheduler may be suboptimal. An intuitive approach is to decode cautiously in the early low-confidence phase and more aggressively in the later steps. **(2) Online Reinforcement Learning**, several recent studies attempt to replicate the success of online reinforcement learning (RL) in AR models to MDLMs. For example, d1 Zhao et al. (2025) directly applies GRPO Shao et al. (2024); Guo et al. (2025) to LLaDA Nie et al. (2025), which overlooks the inconsistency between rollout trajectory and optimization trajectory. While LLaDOU Huang et al. (2025) recognizes this issue and introduces an unmasking policy module to assist low confidence remasking, then combines it with RL to improve performance, this module requires training from scratch and often needs customized training for different model sizes and architectures, which is neither elegant nor scalable. Moreover, even when trajectory consistency considered, hardware memory confronts a significant bottleneck in online reinforcement learning. As maintaining consistency between rollout and optimization trajectories necessitates storing intermediate states throughout the denoising process, memory consumption drastically increases and becomes prohibitive as the number of denoising steps grows.

To this end, we propose approaches from the three perspectives, including decoding strategy, denoising step-size scheduler and reinforcement learning algorithm to alleviate the above problems. In summary, our contributions are three-fold:

- **EOS Early Rejection (EOSER):** We introduce EOS early rejection (EOSER) for full diffusion decoding, which excessively suppress the confidence of `<EOS>` in early steps and gradually restores it as denoising progresses, thereby alleviating the `<EOS>` Trap! . Experimental results show that EOSER significantly improves the performance of full diffusion decoding.

- **Ascending Step-Size (ASS) Decoding Scheduler:** Based on the observation that average token confidence evolves from low to a sharply increasing during denoising, we propose an ascending step-size decoding scheduler instead of uniform step-size ones. This scheduler decodes fewer tokens cautiously in early steps and more tokens aggressively in later steps, substantially reducing the number of decoding steps (e.g., from $\frac{L}{2}$ to $\log_2 L$).

- **Consistency Trajectory Group Relative Policy Optimization (CJ-GRPO):** We propose Consistency Trajectory GRPO (CJ-GRPO) to align the denoising trajectories of rollout and

optimization. It considers the non-causal nature of full diffusion decoding and effectively reduces the optimization errors caused by skip-step optimization. When combined with the EOSER and ASS scheduler, CJ-GRPO exhibits the potential to achieve considerable performance with fewer decoding steps.

## 2 RELATED WORK

### 2.1 MASKED DIFFUSION LANGUAGE MODELS

Recently, utilizing diffusion models to construct large language models (DLMs) Nie et al. (2025); Ye et al. (2025) has emerged as a promising trend. Many works Nie et al. (2025); Ye et al. (2025); Austin et al. (2021); Wu et al. (2023); Sahoo et al. (2024); Lou et al. (2023); Zheng et al. (2024); Gong et al. (2024); Ou et al. (2024); Nie et al. (2024); Arriola et al. (2025) are obsessed with the parallel decoding, flexible order, and potential fewer inference steps of DLMs, considering them a strong competitor to autoregressive models. D3PM Austin et al. (2021) initiates the discrete probability diffusion model, proposing four commonly used corruption approaches to damage sentences in the forward process: uniform, absorbing state (mask), discrete Gaussian, and token embedding distance. Among these, masked diffusion language models (MDLMs) have stood out and established a dominant position among the counterparts. Recent research has scale MDLMs to the billion-parameter regime, achieving performance comparable to similarly sized autoregressive LLMs (e.g., SMDM Gong et al. (2024), LLaDA Nie et al. (2025), Dream Ye et al. (2025), DiffuLLaMA Nie et al. (2024)), significantly scaling their capabilities. Other notable works include Diffusion-LM Li et al. (2022) and simple guidance Schiff et al. (2024) explore controllable text generation for DLMs. Block Diffusion Arriola et al. (2025) introduces semi-autoregressive training-inference paradigm, combining intra-block diffusion, inter-block autoregression to enable variable-length generation.

### 2.2 ONLINE REINFORCEMENT LEARNING FOR DIFFUSION LANGUAGE MODELS

Reinforcement learning (RL) has been extensively explored to align autoregressive (AR) LLMs on human preference and enhancing their reasoning capabilities Schulman et al. (2017); Rafailov et al. (2023); Shao et al. (2024). More recently, MDLMs Ye et al. (2024); Huang et al. (2025); Zhu et al. (2025); Zhao et al. (2025) have shown promise in reasoning tasks (e.g., math, game). For instance, DoT Ye et al. (2024) demonstrates that DLMs are effective in producing chain-of-thought reasoning and addressing reasoning tasks. d1 Zhao et al. (2025) directly adopts the representative online reinforcement learning algorithm (i.e., GRPO Shao et al. (2024); Guo et al. (2025)) to adapt MDLMs, while LLaDOU Huang et al. (2025) explores online reinforcement learning algorithms more suitable for DLMs/MDLMs. So far, most policy gradient-based RL algorithms Shao et al. (2024); Guo et al. (2025); Zeng et al. (2025); Yu et al. (2025); Hu et al. (2025) have been tailored for autoregressive LLMs, which perform with causal mask and token-by-token generation. These algorithms are seamlessly compatible with sequential modeling and decision making. However, the non-causal attention nature of MDLMs prevents RL algorithms designed for AR models from being directly applied to MDLMs.

## 3 METHODOLOGY

### 3.1 MASKED DIFFUSION LARGE LANGUAGE MODELS

Similar to typical diffusion models Ho et al. (2020); Song et al. (2020), MDLMs (Austin et al., 2021; Lou et al., 2023; Shi et al., 2024) model a distribution $p_\theta(x_0)$ through a discrete forward and reverse process defined by $q(x_{t+\tau}|x_t)$ and $p_\theta(x_t|x_{t+\tau})$, respectively. The forward process gradually corrupts a sequence $x_0$ by replacing tokens with `<MASK>` according to a time index $t \in [0, 1]$. At $t = 0$, the sequence $x_0$ is clean and unmasked. The intermediate state $x_t$ is partially masked and each token is masked with the probability $t$ or unmasked with $1 - t$ at time tick $t$. The reverse process recovers the original sequence $x_0$ from the fully masked state $x_1$. During the pre-training, the model learns a mask predictor $p_\theta(\cdot|x_t)$ that takes $x_t$ as input and predicts all `<MASK>`. The training objective minimizes the negative evidence lower bound, which upper-bounds the negative log-likelihood of the model distribution:

$$-\mathbb{E}_{p_{\text{data}}(x_0)}\left[\log p_\theta(x_0)\right] \leq \mathcal{L}_\theta, \tag{1}$$

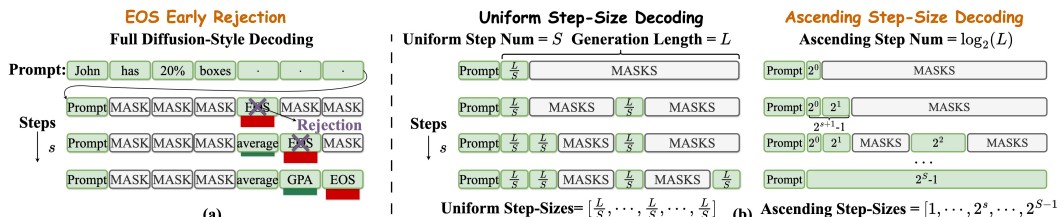

Figure 2: **(a)** EOS Early Rejection. **(b)** Ascending Step-Size Scheduler for fewer inference steps.

$$\mathcal{L}_\theta = -\mathbb{E}_{t\sim\mathcal{U}(0,1),\, x_0\sim p_{\text{data}},\, x_t\sim q_{t|0}(x_t|x_0)} \left[ \frac{1}{t} \sum_i^{|x_t|} \mathbb{I}[x_t^i = \texttt{<MASK>}] \log p_\theta(x_0^i \mid x_t) \right], \qquad (2)$$

where $|x_t|$ is the sequence length of $x$, $k$ means the $k$-th position, $\mathbb{I}$ denotes the indicator function that ensures the loss is computed only for masked tokens. Instruction tuning follows a similar procedure: the prompt $p_0$ remains unchanged, while the response $r_0$ is partially masked. The model takes the prompt $p_0$ and masked response $r_t$ as input and is trained to predict the clean response $r_0$:

$$\mathcal{L}'_\theta = -\mathbb{E}_{t\sim\mathcal{U}(0,1),\, p_0\sim p_{\text{data}},\, r_t\sim q_{t|0}(r_t|p_0,r_0)} \left[ \frac{1}{t} \sum_{i=1}^{|r_t|} \mathbb{I}[r_t^i = \texttt{<MASK>}] \log p_\theta(r_0^i \mid p_0, r_t) \right], \quad (3)$$

where $|r_t|$ is the generation length of response. In practice, the continuous time variable $t \in (0, 1]$ is discretized into the step $\tilde{s}$ with interval $\Delta t = \frac{1}{S}$, such that $t = \tilde{s} \cdot \Delta t, \tilde{s} \in \{0, 1, \cdots, S-1\}$. The denoising step $s$ is defined as $s = S - \tilde{s}, s \in \{1, 0, \cdots, S-1\}$, implying $x_s = x_{S-\tilde{s}} = x_{S-S\cdot t} = x_{1-t}$. For ease of description, we will use $s$ in the remainder of this paper.

## 3.2 EOS EARLY REJECTION

As illustrated in Fig. 1 (Sec. 1), the performance of full diffusion-style decoding significantly lags behind that of semi-autoregressive (block-wise) decoding. This issue arises because LLaDA Nie et al. (2025) replaces `<PAD>` with `<EOS>` during pre-training, resulting the training sequences include a large number of `<EOS>` , and enabling the model to learn such bias from data. Consequently, during the early steps of full diffusion decoding—when the average token confidence is low—the model exhibits disproportionately high confidence in `<EOS>` compared to `<non-EOS>` , leading it prematurely generate `<EOS>` at the end of sequence in early steps. In contrast, commonly used block-wise decoding avoids the `<EOS>` Trap! by isolating later blocks. However, it introduces several limitations: (1) The block boundaries restricts decoding flexibility. Tokens in later blocks that may have high confidence and should be decoded earlier are forcibly delayed, potentially constraining the optimal decoding trajectory. (2) Block length may be a sensitive hyperparameter that requires careful tuning. In comparison, full diffusion-style decoding is free from such constraints.

To alleviate the `<EOS>` Trap! associated with full diffusion-style decoding, we introduce a attenuation coefficient $\gamma$ for `<EOS>` 's confidence that varies with the denoising steps $s$:

$$\gamma = \gamma_{\min} + (\gamma_{\max} - \gamma_{\min})\frac{s}{S-1}, \qquad (4)$$

$$\hat{p}_\theta(x_s^i|x_{s-1}) = \begin{cases} \gamma \cdot p_\theta(x_s^i|x_{s-1}), & \text{if } \text{decode}(x_s^i) = \texttt{EOS} \\ p_\theta(x_s^i|x_{s-1}), & \text{otherwise} \end{cases}, \qquad (5)$$

here, $\gamma_{\min}$ and $\gamma_{\max}$ represent the lower and upper bound, respectively, $s$ is the current step, and $S$ is the total number of steps. The attenuation coefficient $\gamma$ is used for early rejection mechanism. This mechanism suppresses the occurrence of `<EOS>` in the early steps by multiplying the confidence corresponding to early `<EOS>` by the attenuation coefficient $\gamma$, as shown in Eqn. 5 and Fig. 2 (a), while restoring its probability in the later steps to allow for sentence termination. Empirically, we set $\gamma_{max} = 1.0$ and $\gamma_{min} \in [0.4, 0.6]$ for uniform step-size decoding scheduler.

## 3.3 ASCENDING STEP-SIZE SCHEDULER FOR FEWER STEPS INFERENCE

As shown in Fig. 1 and discussed in Sec. 1, the average token confidence remains low in early denoising steps and increases sharply in later steps. This observation suggests that a cautious

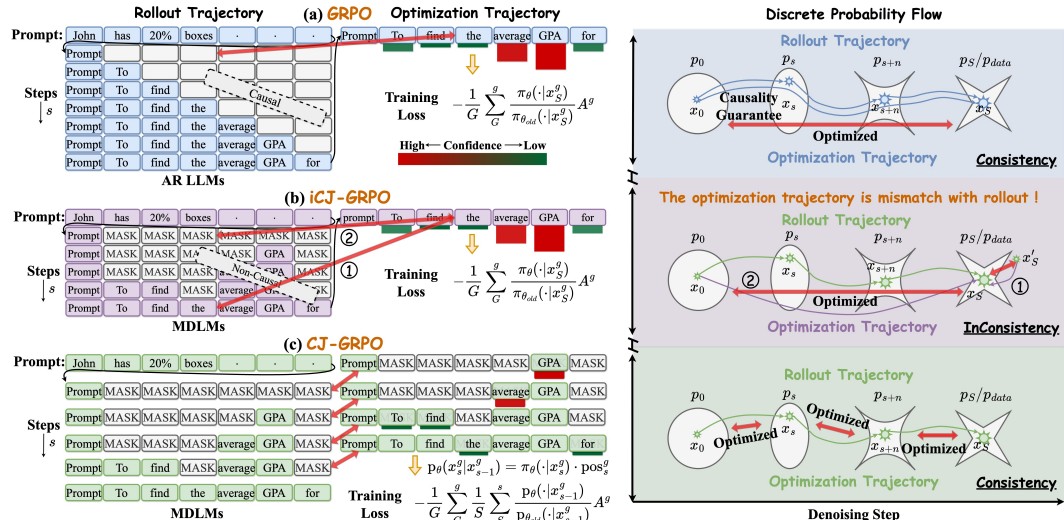

Figure 3: (a) The causality gurarantee the consistency between AR LLMs' rollout and optimization. (b) Two inconsistency trajectories RL training for MDLMs, we refer to ① as one-step $x'_S \to x_S$ and ② as one-step $x_0 \to x_S$. (c) Consistency Trajectory Group Relative Policy Optimization.

decoding strategy—denoising fewer tokens—is preferable initially, while a more aggressive approach—denoising more tokens—can be adopted later. To leverage this phenomenon, we propose an ascending step-size decoding scheduler that adapts to the confidence trend, significantly reducing the number of denoising steps required. As illustrated in Fig. 2 (b), we replace the fixed step-size of $\frac{L}{S}$ with a power-of-2 schedule to determine the number of tokens decoded at each step $s$. Specifically, at decoding step $s \in \{0, \cdots, s, \cdots, S-1\}$, the number of tokens decoded is $2^s$, and the cumulative number of tokens decoded up to step $s$ is $2^{s+1} - 1$. The current and cumulative token counts are thus defined as: $\text{cur\_tokens} = \{1, \cdots, 2^s, \cdots, 2^{S-1}\}$, $\text{cum\_tokens} = \{1, \cdots, 2^{s+1} - 1, \cdots, 2^S - 1\}$. Since the sum of the first S terms of a geometric progression with a common ratio of 2 is $2^S - 1$, but considering that the target generation length $L$ is typically set to a power of 2, we adjust the final step to decode one additional token. Thus the ascending step-size scheduler is defined as follows:

$$\text{Ascending Step-Size Scheduler}(s) = \begin{cases} 2^s + 1, & \text{if } s = S - 1 \\ 2^s, & \text{otherwise} \end{cases}. \tag{6}$$

To avoid the <EOS> Trap! when using Ascending Step-Size (ASS) scheduler with full diffusion decoding, we integrate the EOSER. Given that the step-sizes are non-uniform, we adjust the $\gamma$ to account for the progressive increase in power-of-2. Specifically, the $\gamma_{\text{ASS}}$ is defined as follows:

$$\gamma_{\text{ASS}} = \begin{cases} \gamma_{\min} + (\gamma_{\max} - \gamma_{\min})\dfrac{2^s + 1}{2^S}, & \text{if } s = S - 1 \\ \gamma_{\min} + (\gamma_{\max} - \gamma_{\min})\dfrac{2^s}{2^S}, & \text{otherwise} \end{cases}. \tag{7}$$

Empirically, we set $\gamma_{max} = 1.0$ and $\gamma_{min} = 0.01$. The ASS scheduler can also be extended to block-wise decoding. For example, multiple power-of-2 step groups can be combined into a single block. An example is as follows:

$$\text{Blocks} = \{ \underbrace{2^0, 2^1, 2^2, 2^3, 2^4,}_{\text{block 1, block length}=2^5 - 1} \cdots, \underbrace{2^s, 2^{s+1}, 2^{s+2}, 2^{s+3}, 2^{s+4},}_{\text{block n, block length}=2^s \cdot (2^5 - 1)} \cdots\}. \tag{8}$$

The ASS decoding significantly reduces the inference steps (time) of MDLMs. Several studies Nie et al. (2025); Zhao et al. (2025) demonstrate that the empirically optimal uniform steps for MDLMs are often half the generation length $\frac{L}{2}$. The ASS decoding scheduler reduce this from $\frac{L}{2}$ to $\log_2 L$, thereby lowering the time complexity from $\mathcal{O}(L)$ to $\mathcal{O}(\log_2 L)$. This exhibits significantly potential for accelerating inference.

## 3.4 CONSISTENCY TRAJECTORY GROUP RELATIVE POLICY OPTIMIZATION

GRPO Shao et al. (2024); Guo et al. (2025) is a widely used and cost-effective online reinforcement learning algorithm for training autoregressive (AR) models. However, directly applying GRPO to

masked diffusion language models (MDLMs) may lead to a fundamental mismatch. Unlike AR decoding, Semi-AR and full diffusion-style decoding in MDLM lacks the **causality guarantee** (i.e., causal attention). As depicted in (a) of Fig. 3, in AR models, the predicted confidence of each token depends only on previously decoded tokens, and token confidences obtained during rollout are completely consistent with the confidences computed on final prompt-completion pairs.

In contrast, MDLMs employ bidirectional attention, leading to non-causal (diffusion-style) rollout trajectories. The predicted token confidences during rollout which are estimated in the presence of $\texttt{<MASK>}$ , differ from the confidences computed on final prompt-completion pair ($p_0, r_0$ in Eqn.3). This discrepancy arises because the model's predictions at intermediate steps are conditioned on partially masked sequences, unlike the autoregressive paradigm, as illustrated in Fig. 3 (a). To address the issue posed by non-causal rollout trajectories in MDLMs, there are two compromise **in**consistency tra**j**ectory GRPO algorithm (we refer to them as **iCJ-GRPO**) in Fig.3 (b). ① involves calculating confidence from prompt-completion pairs with slightly perturbed prompts (as in d1 Zhao et al. (2025)). This method can be seen as a one-step denoising optimization from a starting point $x'_S$ near the final state $x_S$, or ② directly computes confidence from fully masked responses $r_1(x_0)$, and optimizing to reach the final answers $r_0(x_S)$, equivalent to performing the one-step denoising optimization from initial state $x_0$ to the final state $x_S$. Both of them may bring significant optimization errors of trajectory and may degrade performance. To ensure consistency between rollout and optimization trajectories, we propose the **C**onsistency Tra**j**ectory **G**roup **R**elative **P**olicy **O**ptimization (**CJ-GRPO**), as illustrated in Fig. 3 (c). During the rollout phase, we record intermediate states at each denoising step $s$ in two queues: one for storing token confidences $\pi_\theta(\cdot|x^g_{s-1})$, and the other for storing the position $\text{pos}^g_s$ of the token decoded at each step, which exhibit the following relationship:

$$p_\theta(x^g_s|x^g_{s-1}) = \pi_\theta(\cdot|x^g_{s-1}) \cdot \text{pos}^g_s, \tag{9}$$

where $p_\theta(x^g_s|x^g_{s-1})$ represents token confidences (probabilities) corresponding to the decoding positions $\text{pos}^g_s$ at step $s-1$, and $g$ denotes the group ID in GRPO. These intermediate states are then utilized to guide the optimization trajectory. The loss between adjacent steps $s-1 \to s$ is as follow:

$$\mathcal{L}_{\theta,s-1} = -\frac{1}{G}\sum_{g=1}^{G}\frac{p_\theta(x^g_s|x^g_{s-1})}{p_{\theta_{\text{old}}}(x^g_s|x^g_{s-1})}A^g + \beta\text{KL}(\pi_\theta\|\pi_{\theta_{\text{ref}}}). \tag{10}$$

The sample-wise trajectory loss and the corresponding batched trajectory loss are:

$$\mathcal{L}^b_\theta = \frac{1}{S}\sum_{s=1}^{S}\mathcal{L}_{\theta,s-1}, \quad \mathcal{L}_\theta = \frac{1}{B}\sum_{b=1}^{B}\mathcal{L}^b_\theta. \tag{11}$$

The detailed optimization pipeline is depicted in Algo. 1. However, a limitation of consistency trajectory GRPO algorithm is that storing intermediate states increases memory overhead proportionally with the number of denoising steps $S$. To address this issue, we naturally consider employing ASS scheduler during rollout. This not only reduces the time complexity from $\mathcal{O}(L)$ to $\mathcal{O}(\log_2 L)$, but also the space complexity from $\mathcal{O}(L)$ to $\mathcal{O}(\log_2 L)$. Furthermore, models trained in this manner exhibit improved fewer-step generation capability. The combination of CJ-GRPO and ASS scheduler brings multiple advantages simultaneously, embodying the *'Kill three birds with one stone'*.

# 4 EXPERIMENTS

Following previous work d1 Zhao et al. (2025), we evaluate LLaDA Nie et al. (2025) after reinforcement learning on mathematical, planning, and coding tasks to assess how our proposed decoding strategies and reinforcement learning algorithm improve reasoning capabilities.

## 4.1 IMPLEMENTATION DETAILS

**Datasets and Benchmarks** We evaluate our method on following domains, (1) **Mathematics**: The mathematical problems which can be verified the correctness between model's response and ground truth answer with robust regular expressions. We utilize GSM8K Cobbe et al. (2021), a high-quality dataset of linguistically diverse grade school math word problems, and MATH500 Lightman et al. (2023), a curated subset of 500 problems filtered from the MATH Hendrycks et al. (2021).

---

**Algorithm 1** CJ-GRPO training with ASS and EOSER

---

**Input** Unmasking policy $\pi_\theta$, training dataset $\mathcal{D}$, reward verifier $R(x, a)$, and queue of denoising trajectory $Q(x_s, \text{pos}_s)$ for recording intermediate states and unmasking positions.

1: **while** $\pi_\theta$ not converged or max epochs unreached **do**
2:      Sample a batch of question-answer pair $(q, a) \in \mathcal{D}$
3:      **for** $g$ **in** $1, ..., G$ **do**              $\triangleright$ $G$ denotes the number of groups per question
4:          Initialize $x_0^g$ with $q$ and mask tokens
5:          **for** $s$ **in** $1, ..., S$ **do**          $\triangleright$ $S$ denotes the number of denoising steps
6:              Decoding $x_s^g \sim \pi_\theta(\cdot|x_{s-1}^g)$ with ASS & EOSER
7:              Enqueue $Q(x_{s-1}^g, \text{pos}_s^g)$
8:          **end for**
9:          $r^g = R(x_S^g, a)$                       $\triangleright$ Compute the rewards
10:      **end for**
11:      $A^g = \frac{r^g - \text{mean}(r^{1:G})}{\text{std}(r^{1:G})}$                    $\triangleright$ Compute the advantages
12:      **for** $i$ **in** $1, ..., \mu$ **do**              $\triangleright$ $\mu$ denotes the number of GRPO iterations
13:          **for** $s$ **in** $1, ..., S$ **do**          $\triangleright$ Compute denoising trajectory loss
14:              Dequeue $Q(x_{s-1}^g, \text{pos}_s^g)$
15:              $p_\theta(x_s^g|x_{s-1}^g) = \pi_\theta(\cdot|x_{s-1}^g) \cdot \text{pos}_s^g, p_{\theta_{\text{old}}}(x_s^g|x_{s-1}^g) = \pi_{\theta_{\text{old}}}(\cdot|x_{s-1}^g) \cdot \text{pos}_s^g$
16:              $\mathcal{L}_{\theta, s-1} = -\frac{1}{G}\sum_{g=1}^{G} \frac{p_\theta(x_s^g|x_{s-1}^g)}{p_{\theta_{\text{old}}}(x_s^g|x_{s-1}^g)} A^g + \beta \text{KL}(\pi_\theta \| \pi_{\theta_{\text{ref}}})$     $\triangleright$ see Eqn. 10
17:          **end for**
18:          $\mathcal{L}_\theta^b = \frac{1}{S}\sum_{s=1}^{S} \mathcal{L}_{\theta,s-1}$, calculate the gradient $\mathcal{L}_\theta = \frac{1}{B}\sum_{b=1}^{B} \mathcal{L}_\theta^b$     $\triangleright$ see Eqn. 11
19:          Update $\theta$ of $\pi_\theta$ with $\nabla_\theta \mathcal{L}_\theta = \frac{1}{B}\sum_{b=1}^{B} \nabla_\theta \mathcal{L}_\theta^b$
20:      **end for**
21: **end while**

**Output** Unmask policy $\pi_\theta$.

---

(2) **Planning**: Involving two categories of tasks: 4x4 Sudoku puzzles, which require constraint satisfaction and systematic elimination to fill a grid with numbers, and Countdown with 3 numbers, a combinatorial arithmetic game where models must reach a target number using basic arithmetic operations on a given set of numbers.

**Model and Training** We utilize LLaDA-8B-Instruct Nie et al. (2025) as the base model. To ensure a fair comparison, we share the same configuration as diffu-GRPO Zhao et al. (2025). For both diffu-GRPO and our CJ-GRPO, the group size for group-relative advantage estimation is 6, the KL-divergence coefficient $\beta$ is 0.04. For supervised fine-tuning (SFT), we train on the s1k Muennighoff et al. (2025) dataset for 20 epochs iteration, and the maximum sequence length of 4096. The importance sampling parameter $\epsilon$ is set to 0.5, and the temperature of Gumbel noise sampling Zheng et al. (2024) during rollout is set to 1.0. The number of GRPO iterations $\mu$ is set to 12 for math tasks and 8 for planning tasks. We set the $\gamma_{max} = 1.0$ for uniform and ASS decoding. For uniform decoding, $\gamma_{min} \in [0.4, 0.6]$, while for ASS scheduler, $\gamma_{min} = 0.01$. All baseline methods (SFT, diffu-GRPO) and CJ-GRPO are trained with a batch size of 48 and 2 gradient accumulation steps.

**Evaluation and metric** We compare our methods against SFT and diffu-GRPO Zhao et al. (2025) across all benchmarks. Evaluation is performed using 0-shot prompting and greedy decoding. The generation lengths are set to 128 and 256, with corresponding block lengths of 64 and 128 (i.e., half the generation length). The number of denoising steps is set to 7/8, 32/64, and 64/128 for the respective configurations.

## 4.2 MAIN RESULTS

We report the performance of baseline LLaDA-8B-Instruct Nie et al. (2025) equipped with different decoding strategies —training-free comparison—in the upper part of Tab. 1 and Tab. 2. The lower sections of the same tables show the results on various post-training recipes across four tasks, evaluated in a zero-shot setting where each model is trained specifically for each task.

**Training-free Setting Comparison Among Decoding Strategies.** We compare the performance of three decoding strategies under the training-free setting: semi-autoregressive (AR), full diffusion and EOSER. For semi-AR decoding, the block length is set to half the generation length, $\frac{L}{2}$, based on empirical studies that show this configuration yields better performance. In contrast, for full diffusion

Table 1: **Performance Comparison on Countdown, GSM8K, MATH500 and Sudoku:** We report results under different decoding strategies and training algorithms, with generation lengths $L$ of 128 and 256 across denoising steps $S=\frac{L}{4}$ and $\frac{L}{2}$ with uniform step-size. The notation '+' indicates a combination of decoding strategies and algorithms, '*' denotes training-free, and gray cell represents our methods. **Bold** font indicates the optimal while underline is the suboptimal performance.

| | Countdown | | | | GSM8K | | | | MATH500 | | | | Sudoku | | | |
|---|---|---|---|---|---|---|---|---|---|---|---|---|---|---|---|---|
| Generation Length $L$ | 128 | | 256 | | 128 | | 256 | | 128 | | 256 | | 128 | | 256 | |
| Methods / Steps $S$ | 32 | 64 | 64 | 128 | 32 | 64 | 64 | 128 | 32 | 64 | 64 | 128 | 32 | 64 | 64 | 128 |
| Semi-AR* | 26.56 | 30.08 | 30.47 | 19.92 | **65.58** | 70.20 | 63.31 | 76.65 | 26.20 | 26.60 | 30.00 | 31.20 | 6.88 | 11.62 | **9.42** | 2.64 |
| Full Diffusion* | 35.94 | 5.08 | 0.00 | 0.00 | 56.94 | 51.86 | 34.87 | 22.97 | 22.20 | 20.40 | 19.60 | 15.60 | 5.76 | 8.98 | 0.00 | 0.83 |
| EOSER* | **44.14** | **46.88** | 41.80 | 39.45 | 60.73 | 63.15 | 58.45 | 61.33 | 22.20 | 24.60 | 22.40 | 24.20 | **13.18** | **13.92** | 2.69 | 1.42 |
| SFT | 23.83 | 30.47 | 18.36 | 24.22 | 61.49 | 68.39 | 61.87 | 77.48 | 23.80 | 27.40 | 23.00 | 31.40 | 6.54 | 8.84 | 11.82 | 4.44 |
| diffu-GRPO | 35.16 | 28.52 | 24.22 | 28.52 | 64.59 | 71.72 | 65.58 | 79.91 | 28.20 | 31.20 | 30.40 | 33.80 | 24.41 | 22.66 | 19.04 | 18.85 |
| CJ-GRPO + Semi-AR | 58.20 | 61.33 | 63.52 | 67.81 | **77.48** | **78.39** | **80.41** | **84.29** | **28.80** | **33.20** | **33.40** | **38.00** | 27.83 | 54.10 | 47.90 | 57.80 |
| CJ-GRPO + EOSER | 66.02 | 70.80 | 69.34 | 75.59 | 62.85 | 67.10 | 71.72 | 77.75 | 23.60 | 23.20 | 26.60 | 28.40 | 85.25 | 85.69 | 85.96 | 85.37 |

Table 2: **Performance of fewer decoding steps on Countdown, GSM8K, MATH500 and Sudoku:** We report results under various decoding strategies and training algorithms when the decoding steps are set to $\log_2 L$. For ascending step-size scheduler, the steps are set to $\log_2 L$, whereas for uniform step-size ones, we fix the steps at 8 to ensure divisibility with both 128 and 256.

| | Countdown | | GSM8K | | MATH500 | | Sudoku | |
|---|---|---|---|---|---|---|---|---|
| Generation Length $L$ | 128 | 256 | 128 | 256 | 128 | 256 | 128 | 256 |
| Methods / Steps $S$ | 7/8 | 8 | 7/8 | 8 | 7/8 | 8 | 7/8 | 8 |
| Semi-AR* + Uniform ($S = 8$) | 18.75 | 4.69 | 42.15 | 26.61 | 17.20 | 13.20 | **6.05** | 0.59 |
| Full Diffusion* + Uniform ($S = 8$) | 11.33 | 0.00 | 44.12 | 12.74 | 17.00 | 9.20 | 0.00 | 0.44 |
| EOSER* + Uniform ($S = 8$) | 5.86 | 22.27 | 43.44 | 29.42 | 16.20 | 14.20 | 0.05 | 0.20 |
| Semi-AR* + ASS ($S = \log_2 L$) | 2.34 | 0.00 | 35.10 | 9.40 | 12.80 | 14.00 | 4.44 | 1.03 |
| Full Diffusion* + ASS ($S = \log_2 L$) | 19.14 | 0.00 | 46.55 | 17.97 | 19.20 | 12.20 | 3.22 | 0.00 |
| EOSER* + ASS ($S = \log_2 L$) | **29.30** | **51.17** | **48.14** | **50.80** | **20.00** | **19.60** | 3.17 | **18.26** |
| SFT + Semi-AR + Uniform ($S = 8$) | 28.91 | 22.66 | 25.25 | 12.28 | 12.00 | 6.40 | 8.64 | 0.73 |
| diffu-GRPO + Semi-AR + Uniform ($S = 8$) | 30.08 | 26.56 | 32.83 | 20.32 | 12.00 | 18.40 | 21.09 | 7.03 |
| CJ-GRPO + Semi-AR + ASS ($S = \log_2 L$) | 3.12 | 1.95 | 38.44 | 5.53 | 16.40 | 19.20 | 24.95 | 19.04 |
| CJ-GRPO + EOSER + ASS ($S = \log_2 L$) | **53.91** | **55.31** | **55.57** | **53.22** | **21.20** | **22.20** | **43.31** | **50.32** |

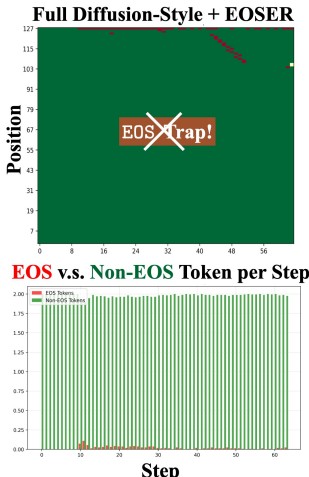

Figure 4: The Step-Position Heatmap of EOSER.

and EOSER, the block length is equal to the generation length, $L$. The results are illustrated in the upper part of Tab. 1 and the Fig. 4. Our findings are as follows: (1) Full diffusion decoding performs significantly worse than the other two strategies across all tasks. In the planning tasks, EOSER outperforms semi-AR, particularly on the Countdown task. However, in mathematics questions, EOSER is inferior to semi-AR. We provide a detailed analysis of this in Sec. 4.4. (2) EOSER effectively mitigates the <EOS> Trap! (i.e., <EOS> are not decoded per step, leading to a large number of <EOS> being ejected in the sequence) and significantly improve the performance of full diffusion. Fig. 4 indicates that the arrival of <EOS> is delayed both spatially (in position) and temporally (in step).

**CJ-GRPO Combined with Semi-AR and EOSER Compared to SFT and diffu-GRPO.** We evaluate the performance of CJ-GRPO combined with semi-AR and EOSER decoding separately, as shown in Tab. 1. We can observe that CJ-GRPO+Semi-AR outperforms both SFT and diffu-GRPO across all tasks and 16 settings. Specifically, for planning tasks, Countdown and Sudoku, CJ-GRPO+Semi-AR achieves performance levels that are twice as high as those of diffu-GRPO in several configurations. Moreover, CJ-GRPO+EOSER significantly surpasses SFT, diffu-GRPO, and even CJ-GRPO+Semi-AR on planning tasks. The improvements are particularly notable on Sudoku, where CJ-GRPO+EOSER achieves performance that is approximately twice that of CJ-GRPO+Semi-AR and four times that of diffu-GRPO. Interestingly, on the mathematics tasks, CJ-GRPO+EOSER underperforms compared to diffu-GRPO and is significantly lower than CJ-GRPO+Semi-AR. This finding is consistent with the results observed in the training-free settings. We provide a detailed analysis of these findings in Sec. 4.4.

**Fewer Decoding Steps Potential of Ascending Step-Size (ASS) Scheduler and CJ-GRPO Combined with ASS.** We evaluate the performance of SFT, diffu-GRPO, and CJ-GRPO using fewer

decoding steps, specifically $\log_2 L$, as depicted in Tab. 2. Under the training-free settings, the combination of EOSER+ASS demonstrates optimal performance across almost all benchmarks on 7 settings, outperforming various decoding methods and step-size schedulers. This result indicates that EOSER+ASS holds significant potential for reducing the number of decoding steps. Next, we train the model using CJ-GRPO combined with EOSER and Semi-AR (note that Semi-AR is also compatible with ASS scheduler as mentioned in Sec. 3.3). We found that even with only 7/8 ($\log_2 L$) steps compared to original 32/64/128 ($\frac{L}{2}$), EOSER+ASS achieves considerable performance, significantly surpassing other methods using either uniform step-size or the ASS scheduler in all settings. Regarding planning tasks, CJ-GRPO+EOSER+ASS even surpasses the original diffu-GRPO which using $\frac{L}{2}$ denoising steps. Please refer to the *Appendix* for more discussion.

## 4.3 ABLATION STUDY

**Ablation on Consistency v.s. Inconsistency in Trajectory Optimization.** We conduct an ablation study to evaluate the two trajectory optimization methods introduced in Sec. 3.4 to clarify the effect of consistent versus inconsistent trajectory optimization. Experiments are conducted with a generation length 128 ($L$), a block length 64 ($\frac{L}{2}$),

Table 3: Ablation on denoising trajectory optimizations.

| | Countdown | GSM8K | MATH500 | Sudoku |
|---|---|---|---|---|
| ① One-step $x'_S$ to $x_S$ | 35.16 | 64.59 | 28.20 | 24.41 |
| ② One-step $x_0$ to $x_S$ | 45.31 | 68.76 | 25.00 | 25.68 |
| Consistency Trajectory $[x_0, \cdots, x_s, \cdots, x_S]$ | **58.20** | **77.48** | **28.80** | **27.83** |

and 32 denoising steps ($\frac{L}{4}$). We refer to ① as one-step from $x'_S$ to $x_S$ (prompt masking as used in d1 Zhao et al. (2025)), and ② as one-step from $x_0$ to $x_S$, and ours as consistency trajectory optimization over $[x_0, \cdots, x_s, \cdots, x_S]$. Results show that optimizing over each intermediate denoising step leads to significantly better performance. Skipping intermediate steps and implementing ② introduces substantial optimization errors of the actual rollout trajectory which leads to optimization biases to some extent. Moreover, implementing ② is equivalent to optimize from a point $x'_S$ near the final solution $x_S$, represents a compromised alternative and yields the worst performance.

## 4.4 ANALYSIS

**Analyzing of Characteristics of MDLMs on Reasoning Tasks.** As noted in Sec. 4.2, training-free and training-based (CJ-GRPO) EOSER decoding significantly outperforms Semi-AR decoding on planning tasks, yet slightly underperforms on mathematical tasks. We attribute this discrepancy to the inherent pattern differences between two types of reasoning tasks, namely planning and math. Specifically, Countdown and planning exhibit a clear bidirectional reasoning pattern, while math have unidirectional pattern. For example, Countdown requires inferring missing operators sandwiched between given operands and target values, while Sudoku demands reasoning over both row and column constraints (2D grid), or thinking forward and backward from the unfilled space (1D sequence) simultaneously. In contrast, math problems in GSM8k and MATH500 typically follow a sequential, unidirectional reasoning process, where each step builds directly on the previous one. This pattern makes semi-AR decoding more suitable for mathematics. These observations suggest the value of future work that hybridizes diffusion-style (bidirectional) and autoregressive (unidirectional) reasoning to better adapt MDLMs to tasks with diverse reasoning patterns.

## 5 CONCLUSION

This paper introduces a reinforcement learning algorithm for MDLMS combined with a full diffusion-style decoding strategy and variable step-size scheduler. The goal is to tame MDLMs via reinforcement learning with fewer decoding steps. We propose three key components: (1) EOS Early Rejection (EOSER), which alleviates the `<EOS>` Trap! in full diffusion-style decoding by suppressing early `<EOS>` generation, (2) Ascending Step-Size (ASS) Decoding Scheduler, which reduces decoding steps while maintaining performance by adapting to the change trend of token confidence—low in early steps and sharply increasing later, and (3) Consistency Trajectory Group Relative Policy Optimization (CJ-GRPO), which aligns rollout and optimization trajectories to ensure the consistency. Experimental results on reasoning tasks demonstrate that CJ-GRPO is particularly effective and efficient for taming MDLMs. When combined with EOSER decoding and ASS scheduler, it achieves comparable performance to $\frac{L}{2}$ steps using only $\log_2 L$ steps.

## 6 ETHICS STATEMENT

Our research focuses on reinforcement learning and decoding strategies for diffusion large language models. We do not involve human subjects in our study. The datasets used in our research are publicly available and do not contain any sensitive or private information. We have ensured that our research practices comply with all relevant legal regulations. We have no conflicts of interest or sponsorship to disclose. Our research aims to improve the performance and efficiency of diffusion large language models, and we believe that our methods and findings are ethical and do not raise any concerns regarding discrimination, bias, fairness, privacy, security, or research integrity.

## 7 REPRODUCIBILITY STATEMENT

Ensuring the reproducibility of our work is a priority. In the experimental section of our paper, we provide detailed descriptions of the parameter settings and experimental configurations for the reinforcement learning and decoding strategies applied to masked diffusion large language models—LLaDA-8B-Instruct. To further support reproducibility, we have open-sourced our codebase to the community. This codebase includes all necessary scripts and instructions for reproducing our results. Additionally, any datasets used in our experiments are publicly accessible. We believe that these efforts will enable other researchers to replicate our findings and build upon our work.

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

## 8 APPENDIX

**Roadmap**

In the Appendix, we present the supplementray experiments and discussions in App. A, the limitations in App. B, and the use of Large Language Models (LLMs) statement in App. C.

## A SUPPLEMENTARY EXPERIMENTS AND DISCUSSIONS

Table 4: **Ablation study on the denoising steps: Countdown.** We report the performance changes as denoising steps varies from 1 to 256, $2^s$, across decoding strategies. '*' denotes training-free.

| Gen Len $L = 256$ | Countdown | | | | | | | | |
|---|---|---|---|---|---|---|---|---|---|
| Denoising Steps | 1 | 2 | 4 | 8 | 16 | 32 | 64 | 128 | 256 |
| Semi-AR* (Block Len $N = 64$) | - | - | 0.00 | 4.69 | 23.83 | 16.02 | 15.62 | **27.34** | 21.09 |
| Full-Diffusion* | 0.00 | 0.00 | 0.00 | 0.00 | **0.78** | 0.00 | 0.00 | 0.00 | 0.00 |
| EOSER* | 0.00 | 0.00 | 8.98 | 22.27 | 21.48 | 23.83 | 38.67 | **41.80** | 39.45 |

Table 5: **Ablation study on the denoising steps: GSM8K.** We report the performance changes as denoising steps varies from 1 to 256, $2^s$, across decoding strategies. '*' denotes training-free.

| Gen Len $L = 256$ | GSM8K | | | | | | | | |
|---|---|---|---|---|---|---|---|---|---|
| Denoising Steps | 1 | 2 | 4 | 8 | 16 | 32 | 64 | 128 | 256 |
| Semi-AR* (Block Len $N = 64$) | - | - | 16.76 | 26.61 | 30.78 | 48.14 | 71.27 | 77.71 | **79.98** |
| Full-Diffusion* | 3.11 | 3.49 | 3.94 | 12.74 | 36.85 | **42.46** | 34.34 | 34.87 | 22.97 |
| EOSER* | 3.11 | 16.22 | 23.28 | 29.42 | 38.59 | 47.76 | 52.08 | 58.45 | **61.33** |

Table 6: **Ablation study on the denoising steps: MATH500.** We report the performance changes as denoising steps varies from 1 to 256, $2^s$, across decoding strategies. '*' denotes training-free.

| Gen Len $L = 256$ | MATH500 | | | | | | | | |
|---|---|---|---|---|---|---|---|---|---|
| Denoising Steps | 1 | 2 | 4 | 8 | 16 | 32 | 64 | 128 | 256 |
| Semi-AR* (Block Len $N = 64$) | - | - | 6.40 | 13.20 | 18.60 | 22.40 | 27.60 | 33.20 | **37.80** |
| Full-Diffusion* | 6.00 | 6.00 | 6.40 | 9.20 | 16.40 | 19.80 | **21.20** | 17.80 | 17.20 |
| EOSER* | 6.00 | 9.60 | 12.20 | 14.20 | 18.20 | 20.20 | 22.40 | **24.20** | 22.67 |

Table 7: **Ablation study on the denoising steps: Sudoku.** We report the performance changes as denoising steps varies from 1 to 256, $2^s$, across decoding strategies. '*' denotes training-free.

| Gen Len $L = 256$ | Sudoku | | | | | | | | |
|---|---|---|---|---|---|---|---|---|---|
| Denoising Steps | 1 | 2 | 4 | 8 | 16 | 32 | 64 | 128 | 256 |
| Semi-AR* (Block Len $N = 64$) | - | - | 0.05 | 0.59 | 2.44 | 5.22 | **10.84** | 5.42 | 5.66 |
| Full-Diffusion* | 0.00 | 0.00 | 0.00 | 0.44 | 8.20 | **8.54** | 3.96 | 0.68 | 0.44 |
| EOSER* | 0.00 | 0.15 | 0.15 | 0.20 | 0.10 | 2.10 | 2.64 | **2.69** | 1.42 |

**Ablation on Denoising Steps.** As depicted in Tab. 4, 5, 6, 7. We conduct ablation studies on the number steps $S$ using different decoding strategies across four benchmarks. We observe a common

trend: for Semi-AR and EOSER, the best performance is often achieved when the number of denoising steps is half the generation length $\frac{L}{2}$. Regarding the full diffusion-style decoding strategy, the optimal number of steps is usually smaller than $\frac{L}{2}$, additionally, while for EOSER (also full diffusion-style), although its optimal number of step is $\frac{L}{2}$, the performance difference across denoising steps such as 64, 128, and 256 is relatively small, however, for Semi-AR decoding strategy, block length may be a sensitive hyperparameter, which may result in a large performance variance obtained. This indicates that full diffusion-style decoding has greater potential for fast generation (fewer decoding steps). Specifically, when the generation length is $L$ and the number of denoising step $\frac{L}{2}$, two tokens are generated at each decoding step. This does not significantly increase the token throughput compared to the next-token prediction of AR models (one-by-one). Moreover, the presence of fixed-length <MASK> means that the generation speed of MDLMs with more decoding steps (e.g. uniform step-size scheduler) may not surpass that of AR models with the same software and hardware infrastructure. Our findings indicate that exploring few-step generation for full diffusion-style decoding in MDLMs is meaningful. Semi-AR, while effective, is not the ultimate solution. Finally, this exploration may truly reveal the speed advantages of MDLMs compared to AR LLMs. Currently, the performance improvements are not significant, and the speed advantages are not fully realized.

## B    LIMITATIONS

In Sec.3.4, we discussed a limitation faced by CJ-GRPO: as the number of steps increases, so does the amount of memory required, because storing intermediate states increases memory overhead proportionally with the number of denoising steps $S$. This constraint greatly affects the training speed. To mitigate this issue, we already have proposed the Ascending Step Size (ASS) scheduler. In planning tasks, the combination of CJ-GRPO with EOSER and ASS, which generates fewer steps ($\log_2 L$), has proven to be even more effective than the previous uniform step-size method that relies on more denoising steps ($\frac{L}{2}$). This not only reduces the time complexity from $\mathcal{O}(L)$ to $\mathcal{O}(\log_2 L)$, but also the space complexity from $\mathcal{O}(L)$ to $\mathcal{O}(\log_2 L)$. However, when it comes to mathematical tasks, this approach still falls short compared to diffu GRPO+Semi-AR and CJ-GRPO+Semi-AR. Given these observations, we believe that exploring better fewer-step full diffusion-style decoding strategies in the future holds great promise. Such methods could potentially resolve the memory issues associated with CJ-GRPO and enhance its mathematical reasoning capabilities. This area of research is well worth exploring.

## C    THE USE OF LARGE LANGUAGE MODELS (LLMS) STATEMENT

In the process of drafting this paper, we utilized large language models (LLMs) as a general-purpose assist tool. Specifically, the LLM was employed to enhance the clarity and coherence of the content, refine the logical structure of the arguments, and improve the grammatical accuracy of the text. The LLM provided suggestions and revisions that helped in polishing the overall quality of the manuscript. However, it is important to note that the LLM did not play a significant role in the research ideation or the core writing of the scientific content. The ideas, methodologies, and conclusions presented in this paper are the original work of the authors. We take full responsibility for the content of this paper, including any text that has been refined using the LLM.

