# OpenReview forum: "Taming Masked Diffusion Language Models via Consistency Trajectory Reinforcement Learning with Fewer Decoding Step"
_ICLR.cc/2026/Conference — ICLR 2026 Conference Withdrawn Submission_

### Official Review · Reviewer_qXYT · 2025-10-23

**Soundness:** 4
**Presentation:** 1
**Contribution:** 1
**Rating:** 2
**Confidence:** 4

**Summary:**

This paper addresses two key challenges in Masked diffusion language models (MDLMs): the "EOS Trap" in full diffusion decoding and training-inference inconsistency in RL for MDLMs. It proposes three components: EOS Early Rejection (EOSER) to suppress premature <EOS> generation, Ascending Step-Size (ASS) decoding scheduler that adapts to token confidence trends to reduce steps from L/2 to log₂L, and Consistency Trajectory Group Relative Policy Optimization (CJ-GRPO) to align rollout and optimization trajectories. Experiments on reasoning tasks (math, planning) with LLaDA-8B-Instruct show that the combination of EOSER, ASS, and CJ-GRPO achieves competitive performance with fewer decoding steps.

**Strengths:**

- Addresses two key issues of Masked Diffusion Language Models (MDLMs) – the "EOS Trap" in full diffusion decoding and training-inference inconsistency in reinforcement learning (RL) for MDLMs with targeted schemes.
- The proposed Ascending Step-Size (ASS) decoding reduces the number of steps from L/2 to log₂L, lowering the time complexity from O(L) to O(log₂L).
- The combination of EOS Early Rejection (EOSER), ASS decoding, and Consistency Trajectory Group Relative Policy Optimization (CJ-GRPO) achieves competitive performance with fewer decoding steps on reasoning tasks such as mathematics and planning.

**Weaknesses:**

1. Lack of training–inference alignment:


Block-wise and semi-autoregressive (semi-AR) decoding strategies are not employed during the training of MDLMs. It is therefore unclear why these strategies outperform full diffusion-style decoding during inference. What is the theoretical or empirical connection between the training objective and the observed inference behavior? Moreover, Table 2 does not seem to show any clear improvement over the full diffusion baseline.

2. Questionable definition of “training–inference inconsistency”:


The paper describes the difference between causal (AR) and non-causal (parallel) decoding as a form of “training–inference inconsistency.” However, this difference appears to be more of a characteristic discrepancy between two decoding paradigms rather than a genuine inconsistency. Why must the decoding process be causal? What concrete problems arise from using non-causal decoding? A deeper analysis or motivation is needed here.

3. Lack of connection and sufficient ablations among proposed techniques:


The three proposed techniques seem loosely related, each targeting a different phenomenon, rather than forming a coherent framework. The claimed effectiveness therefore remains unconvincing. For example, in Table 2, the EOSER component shows only marginal or no advantage even compared to the full diffusion approach, and the RL algorithm is presented without any ablation or justification.

4. Missing comparison with existing methods:


The paper claims that RL algorithms tailored for MDLMs remain underexplored. However,  approach such as LLaDOU already exist, and no comparison with LLaDOU or similar methods is provided, making it hard to position the contribution.

5. Hyperparameter sensitivity and robustness:


In Section 3.2, the paper states that the block length may be a sensitive hyperparameter requiring careful tuning. However, the proposed EOS Early Rejection mechanism introduces even more parameters and tuning complexity, without evidence that it improves robustness or stability.

**Questions:**

Please see questions in Section Weaknesses.

---

### Official Review · Reviewer_nkpx · 2025-10-23

**Soundness:** 2
**Presentation:** 2
**Contribution:** 2
**Rating:** 2
**Confidence:** 4

**Summary:**

This paper focuses on decoding strategies and reinforcement learning algorithms for Masked Diffusion Language Models (MDLMs). The authors observe that MDLMs often generate the end-of-sequence (EOS) token prematurely during early denoising steps and exhibit low confidence when predicting tokens at these stages.

To address these issues, they propose two decoding improvements: EOS Early Rejection (EOSER): suppresses the probability of generating the EOS token during early steps; Ascending Step Size (ASS) scheduler: increases the number of unmasked tokens exponentially across steps.  This allocates more decoding effort to later, cleaner stages and reduces the total denoising complexity from $O(L)$ to $O(\log L)$, where $L$ is the sequence length.

For reinforcement learning, the authors argue that standard GRPO, originally designed for autoregressive language models, cannot be directly applied to MDLMs due to an inconsistency between the rollout and optimization trajectories. To overcome this, they propose Consistency-Trajectory GRPO (CJ-GRPO), which records the token positions unmasked at each denoising step. This enables them to factorize  $ \pi_\theta(\text{completion} \mid \text{prompt} )$ along the observed diffusion trajectory, allowing the GRPO objective to be applied within the MDLM framework.

**Strengths:**

The paper is well motivated. Figure 1c clearly demonstrates the problem of over confidence in EOS and under confidence in non-EOS and hence motivating the EOSER and ASS scheduler.

The evaluation is extensive. The framework is evaluated on Countdown, GSM8K, MATH500 and Sudoku and an ablation study is included.

**Weaknesses:**

In Section 3.2, the authors claimed
> block length may be a sensitive hyperparameter that requries careful tuning. In comparison, full diffusion-style decoding is free from such constraints.

but in equation 4, $\gamma_\min$ and $\gamma_\max$ are also hyperparameters, whose values are determined empirically. I think more clarification is needed to support the claim that tuning block length is a careful work but tuning $\gamma_\min$ and $\gamma_\max$ is not.

In Section 3.4, the authors claimed
>  Unlike AR decoding, Semi-AR and full diffusion-style decoding in MDLM lacks the causality guarantee (i.e., causal attention). ... In contrast, MDLMs employ bidirectional attention, leading to non-causal (diffusion-style) rollout
trajectories.

I don't think the fundamental reason GRPO cannot be naively applied to MDLM is due to the attention pattern but rather the tractability of  $ \pi_\theta(\text{completion} \mid \text{prompt} )$. A clarification on this part is needed.

In the lower half of Table 1, diffu-GRPO is used as a baseline to compare to CJ-GRPO +semi AR or EOSER. This doesn't seem like a fair comparison or at least we cannot disentangle the effect of the decoding strategies from this comparison to evaluate CJ-GRPO.

**Questions:**

Please see Weakness.

---

### Official Review · Reviewer_BepC · 2025-10-31

**Soundness:** 2
**Presentation:** 2
**Contribution:** 2
**Rating:** 4
**Confidence:** 5

**Summary:**

The paper explores a novel approach to improve masked diffusion language models by employing consistency trajectory reinforcement learning. It aims to address the challenges of high computational costs and suboptimal performance in decoding steps. By integrating reinforcement learning techniques that focus on consistency in the trajectory of model outputs, the authors demonstrate a more efficient and effective way to enhance the performance of these models with fewer decoding steps, potentially leading to better scalability and practicality for various natural language processing tasks.

**Strengths:**

- The paper tackles the important problem of few-step fast sampling in diffusion language models, which is a significant challenge in the field.
- The writing is clear and easy to understand. The authors effectively communicate their ideas, methods, and results.

**Weaknesses:**

- Scalability of CJ-GRPO: The paper correctly identifies a major limitation of the proposed CJ-GRPO algorithm: "Storing intermediate states increases memory overhead proportionally with the number of denoising steps S." The paper suggests that the ASS scheduler (which reduces $S$ to $log_{2}L$) mitigates this. However, this solution seems to limit the method's applicability. This algorithm appears difficult to scale to modern reinforcement learning scenarios involving long Chain-of-Thought (CoT) or long-context tasks (e.g., sequence lengths of 8k-32k), as it is only demonstrated on very small-scale tasks (sequence lengths of 128 and 256).

- Missing Comparison with Recent Related Work: The paper's motivation is to improve the efficiency (i.e., reduce steps) of diffusion language models. However, it omits discussion and experimental comparison with other recent and mainstream methods for accelerating dLLM sampling. For instance, [1] and [2] that leverage KV caching or advanced parallel decoding strategies have become critical for making dLLMs competitive. It would be better if the author could make a comparison.


[1] dllm-cache: Accelerating diffusion large language models with adaptive caching, 2506.06295

[2] Fast-dLLM: Training-free Acceleration of Diffusion LLM by Enabling KV Cache and Parallel Decoding, arXiv:2505.22618

**Questions:**

Regarding the $O(S)$ memory complexity of CJ-GRPO: How do the authors envision this method scaling to long-context tasks (e.g., 8k or 32k tokens) that are now common in RLVR? The paper's own results show that on math tasks, reducing $S$ via the ASS scheduler leads to a severe performance drop. This suggests that for complex tasks, a small $S$ is insufficient. Does this not imply that CJ-GRPO is fundamentally limited to short-context tasks where a $log_{2}L$ step count is viable?

---

### Official Review · Reviewer_KQRZ · 2025-11-01

**Soundness:** 3
**Presentation:** 3
**Contribution:** 3
**Rating:** 4
**Confidence:** 3

**Summary:**

This paper studies decoding and online RL for MLM-based dLLMs. It argues that porting AR-ish decoding/likelihood estimation process to such dLLMs is sub-optimal. It then tries to tackle the problem in three folds: 1) the EOS Early Rejection is proposed to avoid getting trapped in early steps; 2) the Ascending Step-Size decoding scheduler is proposed to actually match the theoretical logarithmic performance upperbound in dLLMs' empirical practice; 3) Consistency-traJectory GRPO aligns rollout and optimization trajectories by using per-step states. Experiments confirm the effectiveness of the proposed method.

**Strengths:**

1) It addresses an important problem - the gap between actual decoding procedure and likelihood-estimation (thus RL) training.

2) The proposed methods EOSER, ASS are clear, intuitive methods that are easy to follow.

3) The analysis of how full diffusion variant falls short due to EOS traps is well conducted.

**Weaknesses:**

1) The introduction of EOS decay rate \gamma feels somewhat ad-hoc here. Risk of length bias or delayed termination on tasks that genuinely require short outputs.

2) I respecfully challenge the authors on the novelty of the ASS scheduler. In fact the dLLMs were originally proposed to perform parallel decoding, and doubling the context length is, without a doubt, one of the designed behaviors too. In fact, for insertion-based dLMs like Insertion Transformer, Levenshtein Transformer, it's long been a standard practice and expectation of the logarithmic complexity in decoding. For MLM-style dLLMs, in my opinion people stick with block-wise decoding only because of its generality of applicability. The selected tasks are quite restricted in domains, I don't think it's sufficient to verify the generality of the proposed method.

3) I respecfully beg the authors to reconsider the naming of their approach. "ASS" can be considered really inappropriate for people who'd like to publicly refer to the paper and may affect the paper's publicity.



There are also quite some minor typos (e.g. in caption of Fig. 3. L233 "gurarantee"; "supplementray" L652) and other grammar issues, I'd suggest the authors to spend more time polishing the writing.

**Questions:**

1) Is there any sensitivity analysis available to \gamma?

2) Could you provide some further information on how memory/compute-efficient are the proposed methods?

---

### Note · Authors · 2025-11-24

I have read and agree with the venue's withdrawal policy on behalf of myself and my co-authors.